# End-to-End Vision Tokenizer Tuning

**Wenxuan Wang**[1,2,3*]**, Fan Zhang**[3*]**, Yufeng Cui**[3*]**, Haiwen Diao**[4,3*]**,**
**Zhuoyan Luo**[5,3]**, Huchuan Lu**[4]**, Jing Liu**[1,2†]**, Xinlong Wang**[3†]
[1]Institute of Automation, Chinese Academy of Sciences
[2]School of Artificial Intelligence, University of Chinese Academy of Sciences
[3]Beijing Academy of Artificial Intelligence
[4]Dalian University of Technology [5] Tsinghua University
{wangwenxuan2023@ia.ac.cn,zhangfan@baai.ac.cn,wangxinlong@baai.ac.cn}

## Abstract

Existing vision tokenization isolates the optimization of vision tokenizers from downstream training, implicitly assuming the visual tokens can generalize across various tasks, *e.g.*, image generation and visual question answering. The vision tokenizer optimized for low-level reconstruction is agnostic to downstream tasks requiring varied representations and semantics. This decoupled paradigm introduces a critical misalignment: The loss of the vision tokenization can be the representation bottleneck for target tasks. For example, errors in tokenizing text in an image lead to poor results when recognizing or generating them. To address this, we propose **ETT**, an end-to-end vision tokenizer tuning approach that enables joint optimization between vision tokenization and target autoregressive tasks. Unlike prior autoregressive models that use only discrete indices from a frozen vision tokenizer, **ETT** leverages the visual embeddings of the tokenizer codebook, and optimizes the vision tokenizers end-to-end with both reconstruction and caption objectives. Our **ETT** is simple to implement and integrate, without the need to adjust the original codebooks or architectures of large language models. Extensive experiments demonstrate that our end-to-end vision tokenizer tuning unlocks significant performance gains, *i.e.*, 2-6% for multimodal understanding and visual generation tasks compared to frozen tokenizer baselines, while preserving the original reconstruction capability. We hope this very simple and strong method can empower multimodal foundation models besides image generation and understanding.

## 1 Introduction

Recently, the rapid advancement of large language models (LLMs) and multimodal pre-training has propelled autoregressive (AR) modeling beyond its dominance in natural language processing, extending its influence into the vision and multimodal tasks. Under the next-token prediction (NTP) paradigm of autoregressive models, multimodal learning typically encodes multimodal data such as images and text into compact discrete tokens for unified sequence modeling. For example, a recent work Emu3 [54] tokenizes text, images, and video into discrete tokens and performs next-token prediction in a unified token space. Numerous subsequent works [37, 61, 69, 57], have further advanced this direction, achieving improved performance in visual generation and perception. This unified next-token prediction paradigm enables a flexible multimodal learning framework for both training and inference at scale.

Tokenization plays a key role in an autoregressive framework. For modalities such as image and video, training an efficient and general-purpose tokenizer raises significant challenges as the tokenization can

---

*Equal contribution. † Corresponding author.

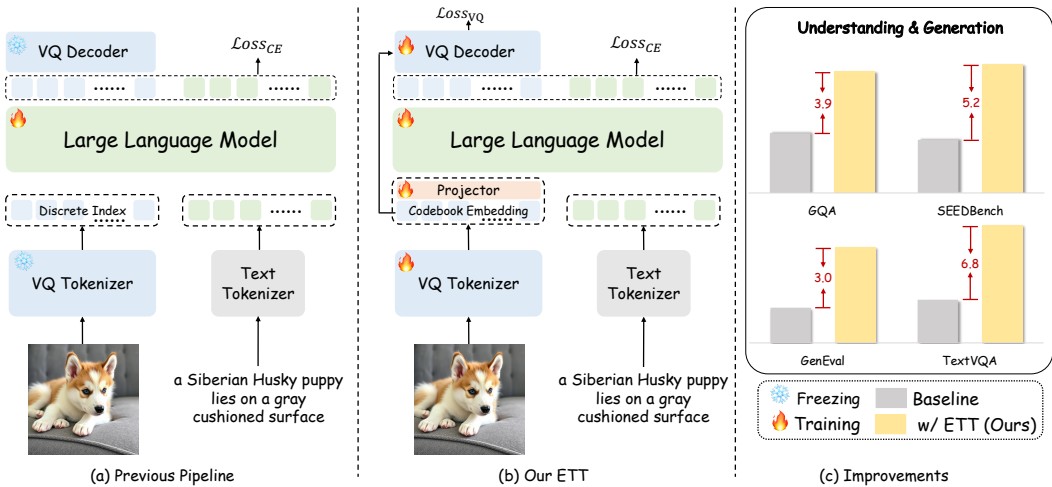

Figure 1: **Left:** Existing autoregressive pipeline uses the discrete indices from a frozen vision tokenizer optimized with low-level reconstruction. **Middle:** We present **ETT**, an end-to-end tokenizer tuning approach which takes advantage of the visual codebook embeddings and optimizes the vision tokenizer and downstream training jointly. **Right:** Our proposed **ETT** unlocks significant performance gains on multimodal understanding and generation benchmarks.

hardly be compact and lossless at the same time. The loss of the tokenization can be the bottleneck for target tasks. For example, errors in tokenizing text in an image lead to poor results when recognizing or generating them. However, existing tokenization approaches neglect this misalignment, and train a vision tokenizer separately and directly integrate it in the downstream training, assuming that the obtained visual tokens can generalize well across tasks. The vision tokenizer optimized for autoencoding is agnostic to the downstream tasks that require various representations and semantics. For instance, most tokenizers focus on low-level pixel-wise reconstruction. The quality of the learned representations is inherently limited by the information loss caused by vector quantization (VQ), leading to inferior performance in visual understanding tasks compared to models using continuous high-level representations like CLIP [38]. Besides, existing autoregressive pipelines typically use only the discrete indices from the vision tokenizer, associated with random initialization of visual embeddings in large language models for realizing downstream tasks. This poses challenges to learn vision representations and vision-language alignment, which are crucial for multimodal learning.

In this work, we present an end-to-end vision tokenizer tuning approach (**ETT**), which enables joint optimization of vision tokenization and target autoregressive downstream tasks. Inspired by recent vision-language models [47, 46, 27, 26] that update their continuous vision encoders during training to optimize the visual representations and vision-language alignment for better visual understanding, we propose to tune the discrete vision tokenizer in an end-to-end manner and leverage large language models as the vision tokenizer's visual assistant. Specifically, we introduce the vision tokenizer's codebook embeddings instead of solely using discrete indices, and integrate token-level caption loss to optimize the representations of the vision tokenizer. Our results highlight that **ETT** significantly boosts the downstream performance of multimodal understanding and visual generation tasks, demonstrating improved discriminative and generative representations in the vision tokenizer. To be noticed, **ETT** can preserve the vision tokenizer's original image reconstruction performance. Our **ETT** is also simple to implement and integrate, without the need to adjust the LLM's original text codebook or expand its embedding layers and classification heads to learn visual embeddings and vision-language alignment from scratch.

Our main contributions can be summarized as follows:

- We present a new vision tokenizer training paradigm to unlock the vision tokenizer for downstream autoregressive tasks. The vision tokenizer is aware of and is optimized for downstream training.
- We introduce a simple yet effective approach **ETT** for end-to-end vision tokenizer tuning. **ETT** leverages the tokenizer's codebook embeddings instead of only using discrete indices, and apply token-level caption loss to optimize the representation of the vision tokenizer.

- **ETT** greatly improves the downstream results in next-token prediction paradigm, for both multi-modal understanding and generation, while maintaining the tokenizer's reconstruction performance.

## 2 Related Work

**Vision Tokenizer**. A vision tokenizer quantizes an image or video into discrete tokens while preserving high reconstruction quality. Specifically, VQ-VAE [53] incorporates a quantizer within an auto-encoder framework, where the quantizer learns to map continuous features into discrete representations. VQGAN [10] improves the reconstruction quality via integrating perceptual loss and adversarial loss. MoVQ [70] proposes to incorporate the spatially conditional normalization to modulate the quantized vectors, generating images with high fidelity. Some other studies focus on improving codebook utilization ratio [71, 63], or incorporating more advanced quantization techniques [68, 35, 65, 25]. Recent works [61, 69, 37, 58] make efforts to integrate semantic information into the tokenizer to improve visual representations. Quantized indices are typically used in downstream tasks, while the tokenizer itself remains frozen during downstream training. In our work, we adopt IBQ [43] as the strong baseline and leverage the codebook embeddings instead of the conventional discrete indices for end-to-end tuning.

**Tokenization for Visual Generation & Understanding.** Discrete visual generation has made great progress in recent years. Some works [40, 64, 45, 62, 21] employ autoregressive approaches to generate images or videos by predicting tokens sequentially. While some others, such as MaskGIT [5] and Muse [4], adopt masked vision token modeling for image generation. Recently, a few studies [54, 48, 59, 58, 55] have explored unifying visual understanding and generation within a single model with the frozen discrete vision tokenizers. For example, Emu3 [54] unifies video, image, and text in token space and achieves strong understanding and generation abilities across modalities via next-token prediction. Show-o [59] proposes to incorporate mask image modeling for image generation within an autoregressive model. Janus [55] introduces two visual encoders for understanding and generation tasks separately to alleviate intrinsic conflicts between two tasks. In this work, we focus on the next-token prediction paradigm and propose to optimize both the vision tokenizer and the autoregressive models jointly for improved visual representation, promoting downstream multimodal generation and perception tasks.

## 3 Methodology

### 3.1 Vision Tokenizer

**Preliminary.** A VQ-based vision tokenizer includes an encoder, a quantizer and a decoder. The encoder projects an input image $I \in \mathbb{R}^{H \times W \times 3}$ to a feature map $f \in \mathbb{R}^{h \times w \times D}$, in which $D$ is the feature dimension and $h = H/s, w = W/s$ with $s$ as the downsampling factor. The quantizer maps each feature in $f$ to the nearest code from codebook $B \in \mathbb{R}^{K \times D}$ to get quantized embeddings $z \in \mathbb{R}^{h \times w \times D}$, where $K$ is the codebook size. Finally, the decoder reconstructs the quantized embeddings back into the original image. The traditional VQ models often struggle in scaling both codebook size and code dimension. As one of the pioneering works, IBQ [43] presents the first attempt to expand the code dimension to 256 while maintaining an extreme large codebook size (*i.e.*, 262,144) by updating the entire codebook simultaneously at each training step.

**Vision Tokenizer in ETT.** We primarily adopt the framework of IBQ [43] for image tokenization, using a downsampling factor of $s = 16$. Each discrete token in codebook has the dimension of $D = 256$. Building upon the original IBQ, we adjust the codebook size to 131,072. The loss function $\mathcal{L}_{vq}$ for tokenizer training is:

$$\mathcal{L}_{vq} = \mathcal{L}_{rec} + \mathcal{L}_{quant} + \mathcal{L}_{lpips} + \lambda_G \cdot \mathcal{L}_{GAN} + \lambda_E \cdot \mathcal{L}_{entropy} \tag{1}$$

where $\mathcal{L}_{rec}$ is the pixel reconstruction loss, $\mathcal{L}_{quant}$ is the quantization loss between quantized embeddings and encoded features, $\mathcal{L}_{lpips}$ is the perceptual loss from LPIPS [67], $\mathcal{L}_{GAN}$ is the adversarial loss from a PatchGAN [18] and $\mathcal{L}_{entropy}$ is the entropy loss [65]. $\lambda_G$ and $\lambda_E$ are the weight of adversarial loss and entropy loss separately.

## 3.2 End-to-End Vision Tokenizer Tuning

**Discrete Indices to Codebook Embeddings.** Methods like Emu3 [54] and similar approaches such as [48], which use only the discrete indices of the vision tokenizer in downstream tasks, discard the rich representational capacity of vision tokenizer embeddings. By depending only on discrete codebook indices, these methods prevent gradient propagation, making end-to-end training infeasible. To address this limitation, we present **ETT**, which directly connects codebook embeddings from vision tokenizer to the LLM, effectively leveraging the richer feature representations encoded within the vision tokenizer while enabling end-to-end training.

**LLM Bridges End-to-End Tuning.** Specifically, as illustrated in Figure 1, given an input image $I \in \mathbb{R}^{H \times W \times 3}$, we first obtain its quantized embeddings $z \in \mathbb{R}^{h \times w \times D}$ from the tokenizer's codebook. To ensure compatibility with the pretrained LLM, we employ a multilayer perceptron with the GeLU activation as a lightweight projector. This projector layer maps the quantized visual embeddings $z$ to $x^I \in \mathbb{R}^{h \times w \times C}$, where $C$ denotes the hidden dimension size of the large language model. Since the entire computation graph, including both the pretrained LLM and the vision tokenizer, remains differentiable, the whole structure can be trained end-to-end using gradient-based optimization. For text input $T$, we utilize the tokenizer and text embedding layer from the pretrained LLM to convert it into text token embeddings $x^T \in \mathbb{R}^{N \times C}$.

**Preservation of Reconstructive Capability.** While end-to-end training enhances the representations of the vision tokenizer, it is essential to maintain its reconstructive capability to ensure high-fidelity image synthesis. To achieve this, we set the overall training objective as the combination of caption loss $\mathcal{L}_{cap}$ and VQ loss $\mathcal{L}_{vq}$. Specifically, we feed both the image token embeddings $x^I$ and text token embeddings $x^T$ into the LLM. For text tokens, we apply the cross-entropy (CE) loss:

$$\mathcal{L}_{cap} = -\sum_{t=1}^{N} \log P(x_t^T | x^I, x_{<t}^T) \tag{2}$$

Besides, we directly reuse the loss function $L_{vq}$ for visual reconstruction. Thus, our end-to-end vision tokenizer tuning objective becomes:

$$\mathcal{L} = \mathcal{L}_{cap} + \alpha \cdot \mathcal{L}_{vq} \tag{3}$$

where $\alpha$ is the loss weight that controls the trade-offs between multimodal perception and visual reconstruction. By jointly training the tokenizer encoder and decoder with LLM, our approach maintains the model's reconstructive capability while ensuring that learned visual tokens remain semantically meaningful and effective for multimodal understanding and generation.

## 3.3 Training Recipe for Multimodal Generation and Understanding

Following previous works [7, 8], the whole training process for downstream multimodal perception and generation follows three sequential training stages. The employed training data comprises publicly available image datasets supplemented with diverse instruction data for understanding and generation, as shown in Table 1.

Table 1: **Details of training data across all stages for multimodal generation and perception.**

| Stage | Dataset | #Num | Total |
|---|---|---|---|
| Stage 1: Alignment Learning | SOL-recap | 12.0M | 12.0M |
| Stage 2: Semantic Learning | SOL-recap | 12.0M | 12.0M |
| Stage 3: Post-Training Chat | SOL-recap | 32.0M | 67.3M |
| | LLaVA-OneVision [27] | 3.5M | |
| | Infinity-MM [15] | 31.8M | |
| Stage 3: Post-Training Gen | AI-generated Data | 14.0M | 30.0M |
| | High-Aesthetics Web Data | 16.0M | |

**Stage 1: Alignment learning.** The first training stage is to effectively establish the vision-language alignment. With the pretrained large language model and vision tokenizer, we keep them frozen and train only the visual projector layer with image-to-text caption loss $\mathcal{L}_{cap}$. This setup enables the

LLM to acquire visual concepts and entities directly from the tokenizer, effectively bridging vision and language modalities. Specifically, we curate a 12M-image subset from our constructed dataset SOL-recap, which comprises 32M image-text pairs sourced from publicly available datasets, *i.e.*, SA-1B [20], OpenImages [22], and LAION [42]. To be noticed, all images are recaptioned using an improved captioning engine following [8]. The high-quality data at this stage can enhance training stability and cross-modality alignment.

**Stage 2: Semantic Learning.** The second stage acts as the most critical part in the entire training pipeline, realizing end-to-end vision tokenizer tuning. At this stage, we unfreeze the weights of the LLM, projector, and vision tokenizer, optimizing them using the caption loss $\mathcal{L}_{cap}$ and the reconstruction loss $\mathcal{L}_{vq}$ jointly, as defined in Equation 3. The high-quality subset, *i.e.*, 12M image-text pairs from SOL-recap, is utilized for multimodal understanding and reconstruction learning. This stage enables efficient learning of the vision tokenizer's perceptual capability, supporting both visual reconstruction and understanding. This well designed stage 2 can enhance alignment between vision tokenizer and downstream tasks while preserving the original reconstructive ability.

**Stage 3: Post-Training.** After acquiring the enhanced vision tokenizer via the proposed end-to-end vision tokenizer tuning, we follow the standard post-training pipeline to realize multimodal understanding and generation. During this training stage, we further post-train two specialist models, *i.e.***ETT-Chat** and **ETT-Gen**, by freezing the vision tokenizer part, and tuning the visual projector, as well as the large language model layers to enhance instruction-following capabilities for multimodal understanding and text-to-image generation, respectively. For multimodal understanding, we collect a diverse set of high-quality, multi-source instruction data, including SOL-recap, LLaVA-OneVision [27], and Infinity-MM [15]. For visual generation, we construct 14M AI-generated samples using the Flux model [23] and additionally curate 16M image-text pairs from open-source web data [12, 3], filtering them based on image resolution and LAION aesthetic score [24].

# 4 Experimental Results

## 4.1 Training Settings

**Data Preparation.** *(1) Vision-Language Pre-training & Vision Tokenizer Datasets.* We adopt the pre-processing pipeline [8] to refine SA-1B [20], OpenImages [22], and LAION [42], resulting in 11M, 7M, and 14M images respectively. We utilize the caption engine following [8] to produce 32M high-quality captions. *(2) Supervised Fine-tuning Datasets.* For understanding datasets, we extract 31.8M multi-task samples from Infinity-MM [15] and 3.5M instruction data from LLaVA-OneVision [27], prioritizing complex conversational structures. For generation datasets, we generate 14M AI-created samples with the Flux model [23] and further select 16M image-text pairs from open-source web data [12, 3], applying filters based on image resolution and aesthetic scores [24].

**Implementation Details.** We train **ETT** on 8-A100 nodes using the Adam optimizer [19]. The batch sizes for Stages 1, 2, and 3 are set to 1024, 1024, and 1024, respectively, with maximum learning rates of $4 \times 10^{-5}$, $4 \times 10^{-5}$, and $2 \times 10^{-5}$. We apply a warm-up strategy with a 0.03 ratio and use a cosine decay scheduler across all stages. Unless otherwise specified, images are processed at a resolution of $512^2$, and ablation studies are reported using LLaVA-mix-665K [31] at Stages 3. For all the experiments in our work, we adopt Qwen2.5-1.5B [50] as the large language model for multimodal sequence modeling.

For the vision tokenizer in **ETT**, we employ Adam optimizer [19] with a fixed learning rate of $1 \times 10^{-4}$, $\beta_1 = 0.5$ and $\beta_2 = 0.9$. The tokenizer is trained for 500,000 steps with a global batch size of 256 and an input resolution of $256 \times 256$. The adversarial loss weight $\lambda_G$ is set to 0.1, and the entropy loss weight $\lambda_E$ is set to 0.05. We also adopt LeCAM regularization [52] for discriminator training to improve training stability.

## 4.2 Multimodal Understanding Evaluation

We validate **ETT** on various widely known vision-language perception benchmarks, covering task-specific evaluations (GQA [17] and TextVQA [44]), hallucination detection (POPE [29]), open-domain multimodal understanding (MME [11], MMBench [33], SEED-Bench [28], and MM-Vet [66]), and scientific reasoning (ScienceQA-IMG [34]).

Table 2: **Comparison with existing state-of-the-art vision-language models on various multi-modal understanding benchmarks**, including MMB[EN]: MMBench-EN [33]; SEED[I]: SEEDBench-Img [28]; MMV: MMVet [66]; MME [11]; POPE [29]; GQA [17]; SQA[I]: ScienceQA-Img [34]; TQA: TextVQA [44]. Note that #LLM-Param denotes the number of LLM parameters, #Data represents the pre-training / fine-tuning data volume, * denotes that the images of related training datasets are observed during training, and the best results are marked in **bold**.

| Method | #LLM-Param | #Data | MMB[en] | SEED[I] | MMV | MME | POPE | GQA | SQA[I] | TQA |
|---|---|---|---|---|---|---|---|---|---|---|
| *Continuous VLMs:* | | | | | | | | | | |
| QwenVL-Chat [1] | 7B | 7.2B / 50M | 60.6 | 58.2 | - | 1848.0 | - | 57.5 | 68.2 | 61.5 |
| EVE [7] | 7B | 33M / 1.8M | 52.3 | 64.6 | 25.7 | 1628.0 | 85.0 | 62.6 | 64.9 | 56.8 |
| Cambrian [51] | 7B | 10B+ / 7M | **75.9** | **74.7** | - | - | - | **64.6** | 80.4 | **71.7** |
| LLaVA-1.5 [31] | 7B | 0.4B+ / 665K | 64.3 | 64.3 | 30.5 | **1859.0** | 85.9 | 62.0 | 66.8 | 46.1 |
| LLaVA-1.6 [32] | 7B | 0.4B+ / 760K | 67.4 | 64.7 | **43.9** | 1842.0 | 86.4 | 64.2 | 70.2 | 64.9 |
| Janus [55] | 1.3B | - / - | 69.4 | 63.7 | 34.3 | 1338.0 | **87.0** | 59.1 | - | - |
| *Discrete VLMs:* | | | | | | | | | | |
| Chameleon [48] | 7B | 1.4B+ / 1.8M | 31.1 | 30.6 | 8.3 | 170.0 | - | - | 47.2 | 4.8 |
| LWM [30] | 7B | 1B+/- | - | - | 9.6 | - | 75.2 | 44.8 | - | - |
| VILA-U [58] | 7B | 700M/7M | - | - | 33.5 | - | 85.8 | 60.8 | - | 60.8 |
| Emu3 [54] | 8B | - / - | 58.5 | 68.2 | 37.2 | - | 85.2 | 60.3 | 89.2* | 64.7 |
| Show-o [59] | 1.3B | 2.0B+/665K | - | - | - | - | 80.0 | 58.0 | - | - |
| Liquid [57] | 7B | 40M / 3.5M | - | - | - | 1119.3 | 81.1 | 58.4 | - | 42.4 |
| **ETT** | 1.5B | 32M / 35.3M | 58.8 | 66.6 | 29.3 | 1532.6 | 82.4 | 59.4 | 91.7* | 56.8 |
| **ETT** | 3B | 32M / 35.3M | 61.3 | 67.8 | 32.2 | 1513.9 | 83.2 | 60.5 | **93.3*** | 61.1 |

As shown in Table 2, our **ETT** consistently outperforms discrete counterparts, such as Chameleon [49], LWM [30], and Liquid [57], even with smaller model and data scales. This highlights the efficacy of **ETT**'s end-to-end tuning strategy, which enables to achieve strong results with fewer parameters and less data. Additionally, **ETT** achieves superior performance compared to Show-o [60] across a wide range of benchmarks, despite being trained on significantly less data. This underscores the effectiveness of **ETT**'s data utilization strategies and its ability to generalize well even with limited training resources. Furthermore, **ETT** demonstrates competitive performance against state-of-the-art (SOTA) continuous encoder-based VLMs, including QwenVL-Chat [1], EVE [7], and Janus [56], without relying on additional visual encoders for complex image encoding tasks. This not only simplifies the model architecture but also reduces computational overhead, making **ETT** a more practical and scalable solution for multimodal tasks. The success of **ETT** lies in its end-to-end training approach for visual tokenization, which seamlessly integrates multimodal understanding and generation while resolving internal conflicts within the tokenizer.

Table 3: **Comparison with existing state-of-the-art vision-language models on various text-to-image generation benchmarks**, including GenEval [14] and T2I-CompBench [16]. Note that #LLM-Param denotes the number of LLM parameters, #Data represents the training data volume, and the best results are marked in **bold**.

| Method | #LLM-Param | #Data | GenEval Overall ↑ | Single ↑ | Two ↑ | Counting ↑ | Colors ↑ | Position ↑ | ColorAttr ↑ | T2I-CompBench Color ↑ | Shape ↑ | Texture ↑ |
|---|---|---|---|---|---|---|---|---|---|---|---|---|
| *Generation on Continuous Features:* | | | | | | | | | | | | |
| SEED-X [13] | 17B | - | 0.51 | 0.96 | 0.65 | 0.31 | 0.80 | 0.18 | 0.14 | - | - | - |
| PixArt-α [6] | 0.6B | 25M | 0.48 | 0.98 | 0.50 | 0.44 | 0.80 | 0.08 | 0.07 | 68.86 | 55.82 | 70.44 |
| SD v1.5 [41] | 1B | 2B | 0.43 | 0.97 | 0.38 | 0.35 | 0.76 | 0.04 | 0.06 | 37.50 | 37.24 | 42.19 |
| SD v2.1 [41] | 1B | 2B | 0.50 | 0.98 | 0.37 | 0.44 | **0.85** | 0.07 | 0.17 | 56.94 | 44.95 | 49.82 |
| SDXL [36] | 2.6B | - | 0.55 | 0.98 | 0.44 | 0.39 | **0.85** | 0.15 | 0.23 | 63.69 | 54.08 | 56.37 |
| DALL-E2 [39] | 6.5B | 650M | 0.52 | 0.94 | 0.66 | 0.49 | 0.77 | 0.10 | 0.19 | 57.50 | 54.64 | 63.74 |
| DALL-E3 [2] | - | - | **0.67** | 0.96 | **0.87** | 0.47 | 0.83 | 0.43 | **0.45** | 81.10 | 67.50 | **80.70** |
| SD3 [9] | 2B | - | 0.62 | 0.98 | 0.74 | **0.63** | 0.67 | 0.34 | 0.36 | - | - | - |
| *Generation on Discrete Features:* | | | | | | | | | | | | |
| LlamaGen [45] | 0.8B | 60M | 0.32 | 0.71 | 0.34 | 0.21 | 0.58 | 0.07 | 0.04 | - | - | - |
| Show-o [59] | 1.3B | 35M | 0.53 | 0.95 | 0.52 | 0.49 | 0.82 | 0.11 | 0.28 | - | - | - |
| LWM [30] | 7B | - | 0.47 | 0.93 | 0.41 | 0.46 | 0.79 | 0.09 | 0.15 | - | - | - |
| Chameleon [48] | 34B | - | 0.39 | - | - | - | - | - | - | - | - | - |
| Emu3 [54] | 8B | - | 0.66 | **0.99** | 0.81 | 0.42 | 0.80 | **0.49** | **0.45** | 79.13 | 58.46 | 74.22 |
| Janus [55] | 1.3B | - | 0.61 | 0.97 | 0.68 | 0.30 | 0.84 | 0.46 | 0.42 | - | - | - |
| **ETT** | 1.5B | 30M | 0.63 | 0.98 | 0.81 | 0.25 | 0.84 | 0.48 | **0.45** | 81.03 | 58.19 | 72.14 |

## 4.3 Visual Generation Evaluation

We comprehensively evaluate the text-to-image generation capabilities of our model against previous diffusion-based and autoregressive-based SOTA methods, including both multimodal specialists and generalists, on widely adopted benchmark datasets GenEval [14] and T2I-CompBench [16]. As demonstrated in Table 3, our method achieves competitive performance while utilizing much fewer LLM parameters and a smaller-scale training dataset. Specifically, under the inference configuration of top-$k$=131,072 (*i.e.*, visual vocabulary size) and top-$p$=1.0, our model attains an overall score of 0.63 on the GenEval dataset, outperforming advanced diffusion models such as SDXL [36]. Furthermore, our approach surpasses autoregressive-based methods, including both specialists (*e.g.*, LlamaGen [45]) and generalists (*e.g.*, Chameleon [48]), requiring either less training data or fewer model parameters. When enhanced with prompt rewriting, the achieved model accuracy by our method closely approaches the performance of current leading models such as DALL-E 3 [2] and EMU3 [54]. On T2I-CompBench dataset [16], our model achieves promising performance with scores of 81.03, 58.19, and 72.14 on color, shape, and texture pattern, respectively, demonstrating competitive performance compared to SOTA diffusion-based counterparts. These results fully prove the efficacy of our end-to-end vision tokenizer tuning approach, highlighting its ability to achieve superior multimodal understanding and generation performance across diverse benchmarks.

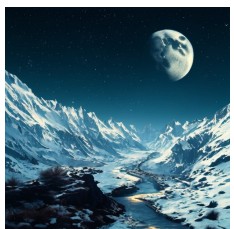
A snowy mountain landscape under a starry sky, with jagged peaks, a reflective river, and a cratered moon, balanced by warm and cool tones.

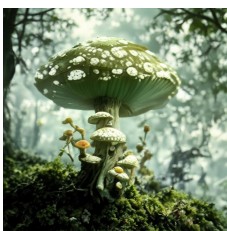
A large, textured green mushroom with white spots grows from a mossy forest floor, framed by misty trees and soft, glowing light.

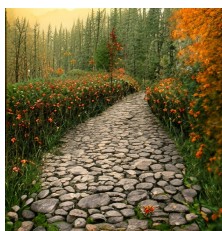
A winding pebble path cuts through a colorful flower field toward a rustic hut, backed by a forest and bathed in warm autumn light, richly detailed in sharp 8K.

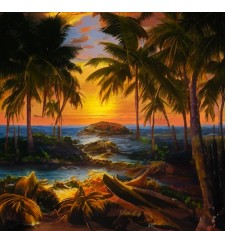
A surreal oil painting of a Sri Lankan sunset, with coconut trees in the foreground and a glowing ocean horizon, rich in volumetric light and shadow.

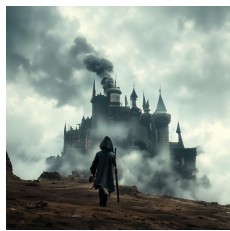
A lone figure walks towards a mist-shrouded, dilapidated castle, surrounded by a barren landscape. The dark, muted tones create a sense of isolation and foreboding.

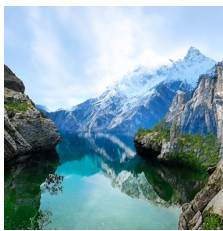
A serene pan across a mountain landscape reveals snow-capped peaks, rugged rocks, and a clear lake reflecting the sky, evoking peace and wonder.

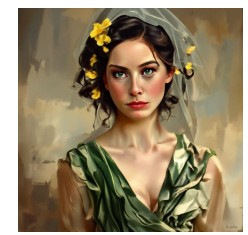
A realistic portrait of a woman with fair skin, soft eyes, and loose curls with yellow flowers, wearing a green and cream dress, with a sheer veil and a muted earth-tone background.

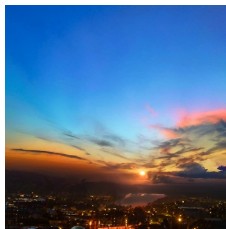
As the sun sets, warm orange light fills the sky, now streaked with pink and purple clouds. Below, the city bustles, traffic lights flickering to life.

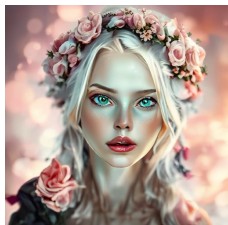
A surreal portrait of a woman with platinum blonde hair and green eyes, framed by a pink rose wreath and soft pastel background.

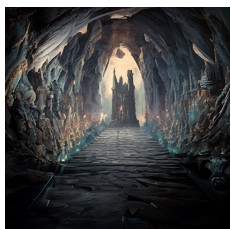
Wide-angle view of a dark fantasy castle, with a torch-lit pathway through an abstract portal, set in a moody, sci-fi fantasy world with 8K detail.

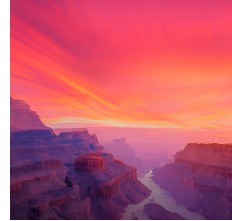
A vivid sunset bathes the Grand Canyon in pink and orange light, silhouetting its towering rocks as the Colorado River winds below.

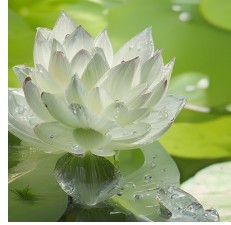
A close-up of a lotus flower formed from crystal-clear water droplets, set against lily pads on a tranquil pond, with sunlight highlighting its shimmering petals.

Figure 2: **Visual generation results with our ETT.** We present 512 × 512 results spanning different styles, subjects, and scenarios. Note that the presented prompts are simplified versions, which convey the general meaning.

Fig. 2 presents qualitative results generated by **ETT**, demonstrating its ability to follow prompts accurately while generating diverse visual content. The model exhibits proficiency in producing images across various artistic styles, subjects, and backgrounds, adapting to different compositional structures and aesthetic preferences.

## 4.4 Ablation Study

To verify the effectiveness of our **ETT** for downstream multimodal generation and understanding tasks, we conduct comprehensive ablation studies on several prevalent understanding benchmarks (*e.g.*, SEEDBench-Img [28], GQA [17], TextVQA [44] and MME-Perception [11]) that assess diverse capabilities, as well as the GenEval dataset [14] for the evaluation of text-to-image generation.

Table 4: **Ablation study on the benefits of ETT for multimodal understanding and generation.**

| Visual | Tuning | | ETT for Und. | | | ETT for Gen. |
| | | $SEED^I$ | GQA | TQA | $MME^P$ | GenEval-O↑ |
|---|---|---|---|---|---|---|
| Index | ✗ | 54.8 | 50.9 | 40.1 | 1028.7 | 0.40 |
| Embed | ✗ | 54.8 | 52.6 | 40.9 | 1034.0 | 0.34 |
| Embed | ✓ | **60.0** | **54.8** | **46.9** | **1124.7** | **0.43** |

**End-to-End Tuning Benefits.** We first probe into the effectiveness of our **ETT** for promoting multimodal downstream tasks. To ensure a fair comparison in validating the potential of **ETT** in optimizing vision tokenizer's feature representations, we train all models for understanding and generation tasks with SOL-recap. Additionally, we apply LLaVA-mix-665K [31] for an extra supervised fine-tuning stage for understanding tasks. As presented in Table 4, introducing **ETT** consistently yields significant performance improvements across both understanding and generation tasks, compared with the traditional tokenizer exploitation manner. Specifically, without end-to-end tuning, replacing the discrete indices with codebook embeddings partially alleviates the issue of information loss and brings notable performance gains on multimodal understanding benchmarks. Although this replacement degrades the visual generation performance, it establishes a fully differentiable model architecture, allowing for end-to-end optimization. Building on this foundation, incorporating end-to-end tuning the visual tokenizer further enhances performance for both understanding and generation tasks compared with the conventional setting (*i.e.*, first row), particularly on tasks that heavily rely on visual features (*e.g.*, ↑ 5% for general visual question answering [28] and ↑ 6% for optical character recognition [44]).

Table 5: **Ablation study on the impact of ETT for the trade-offs between multimodal perception and visual reconstruction.**

| Tuning Tasks for VQ | $\alpha$ | | Understanding | | | | Reconstruction | |
| | | $SEED^I$ | GQA | TQA | $MME^P$ | ImageNet-rFID↓ | ImageNet-PSNR↑ | ImageNet-SSIM↑ |
|---|---|---|---|---|---|---|---|---|
| - | - | 54.8 | 52.6 | 40.9 | 1034.0 | **1.03** | **21.73** | **0.60** |
| Und. only | - | **61.2** | **55.2** | **48.0** | **1164.3** | 45.70 | 13.22 | 0.19 |
| Und.&Rec. | 0.25 | 60.0 | 54.8 | 46.9 | 1124.7 | 1.65 | 20.89 | 0.58 |
| Und.&Rec. | 0.5 | 59.9 | 54.7 | 46.9 | 1118.1 | 1.66 | 21.14 | 0.59 |
| Und.&Rec. | 1.0 | 59.3 | 53.9 | 45.7 | 1088.2 | 1.50 | 21.48 | **0.60** |

**Trade-offs between I-to-T & Reconstruction.** Then, we investigate the inherent task trade-off between visual reconstruction and multimodal understanding of **ETT**. The commonly used reconstruction metrics including rFID, PSNR and SSIM are all adopted for the comprehensive evaluations. As shown in Table 5, compared to untuned baseline (*i.e.*, first row), tuning vision tokenizer consistently delivers substantial gains for the understanding task, albeit at the cost of reconstruction performance, which deteriorates to varying extents. Specifically, tuning the vision tokenizer with only image-to-text understanding task (*i.e.*, second row) yields the best performance on various understanding benchmarks but greatly deteriorate in reconstruction, *e.g.*, the rFID on ImageNet $256 \times 256$ setting dramatically drops from 1.033 to 45.701. Introducing auxiliary reconstruction target with a small weight 0.25 slightly reduces understanding accuracy while significantly improving the reconstruction, indicating the importance of joint training on both understanding and reconstruction tasks. Increasing the reconstruction weight $\alpha$ to 1.0 achieves the best reconstruction accuracy but results in the weakest

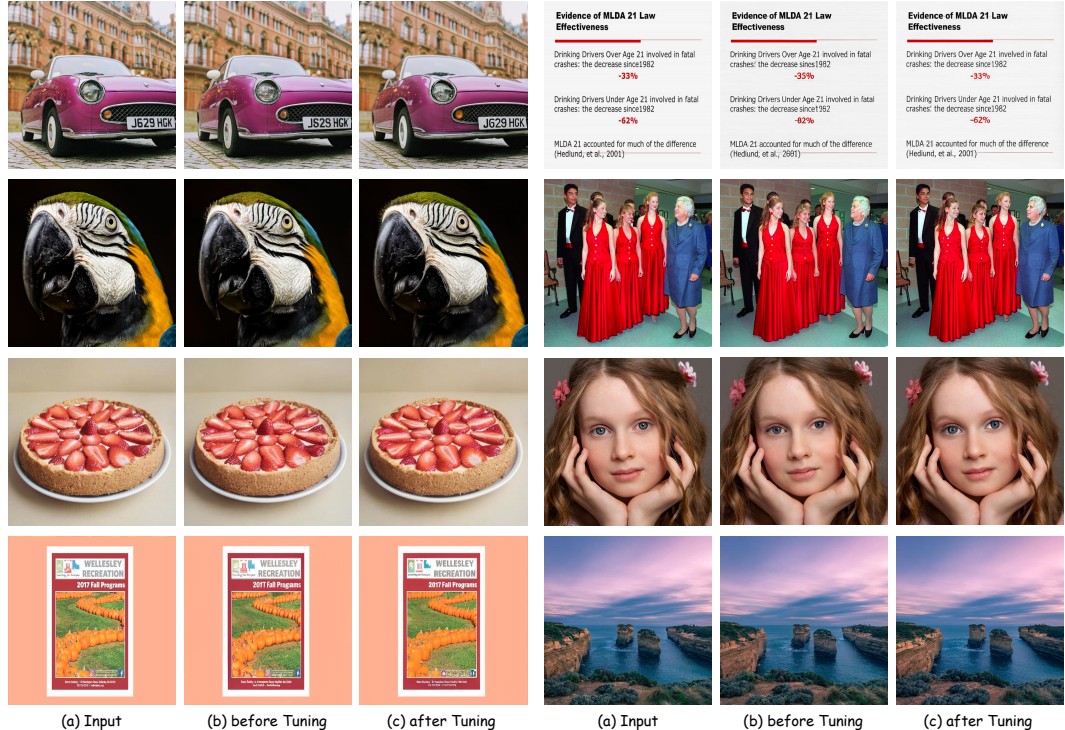

| (a) Input | (b) before Tuning | (c) after Tuning | (a) Input | (b) before Tuning | (c) after Tuning |

Figure 3: **Comparison of visual reconstruction results of the input image before and after end-to-end tuning by our ETT.** The vision tokenizer tuned by our **ETT** benefits from retaining the original rich low-level detail representations while being effectively injected with high-level semantics, producing visual details comparable to the pre-tuned counterpart and even performs better in some detail reconstruction, *e.g.*, text rendering.

perception capability. Therefore, to balance both understanding and reconstruction tasks, we choose 0.25 as the default reconstruction loss weight $\alpha$.

Besides, we also visualize the reconstruction results before and after introducing **ETT** in Figure 3. The vision tokenizer, when tuned with **ETT**, generates visual details comparable to its untuned counterpart while enhancing specific aspects, such as text rendering. This suggests that **ETT** can not only preserve the original rich low-level details but also improve high-level semantic representations.

## 5 Conclusion

In this work, we focus on addressing the representation bottleneck of the vision tokenizer for multimodal learning. We introduce a simple yet effective approach of end-to-end vision tokenizer tuning, termed **ETT**. **ETT** involves codebook embeddings instead of solely discrete indices and applies token-level caption loss for end-to-end optimization of both the tokenizer and downstream training. **ETT** significantly enhances the multimodal understanding and generation with a decoder-only architecture, while almost preserving the tokenizer's reconstruction capability and even boosting reconstruction performance on specific aspects such as text rendering.

## 6 Limitations and Future Topics

One potential limitation of this work is that both the data scale for end-to-end fine-tuning and the model capacity could be further expanded to enhance visual representations and downstream task performance. Moreover, our current approach primarily focuses on designing a simple yet effective framework that optimizes the visual features of existing vision tokenizers, leveraging the semantic capabilities of LLMs, rather than constructing a vision tokenizer inherently designed for

both understanding and generation through unified end-to-end training. While **ETT** shows the potential of using LLM-driven semantic feedback to enhance vision tokenization, it still relies on fine-tuning pre-existing tokenizers rather than developing one from the ground up. Therefore, for future research we will explore end-to-end training of a vision tokenizer from scratch, aiming to create a more comprehensive and adaptable representation for multimodal tasks. Besides, extending beyond image and text modalities, such as incorporating video and audio, presents an exciting avenue for further advancements. We hope that our simple yet effective method can contribute to the broader development of multimodal foundation models beyond visual generation and understanding.

## Acknowledgment

We thank all the insightful reviewers for the helpful suggestions, and the colleagues at Beijing Academy of Artificial Intelligence for support throughout this project. This research is supported by Artificial Intelligence-National Science and Technology Major Project (2023ZD0121200) and the National Natural Science Foundation of China (62437001, 62436001, U21B2043), and the Natural Science Foundation of Jiangsu Province under Grant BK20243051.

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
