# OpenReview forum: "End-to-End Vision Tokenizer Tuning"
_NeurIPS.cc/2025/Conference — NeurIPS 2025 poster_

### Official Review · Reviewer_t4q5 · 2025-06-26

**Clarity:** 3
**Significance:** 3
**Originality:** 2
**Rating:** 5
**Confidence:** 3

**Summary:**

The paper proposes end-to-end tuning of the vision tokenizer to mitigate the misalignment errors steaming from decoupled training of the vision tokenizer and the target Vision-Language (VL) tasks. It proposes to jointly optimize the vision tokenizer and the LLM with the reconstruction and captioning objectives. This is achieved by using the continuous visual embeddings of the tokenizer codebook instead of the discrete indices from the frozen vision tokenizer.

**Questions:**

* As I wrote above, it is hard to judge from where the improvements in Table 2 and 3 come from. Is it because your curated data is of higher quality or is it because of the ETT method or both? I think it would be very valuable to decouple these, for example with an experiment where you train one of the benchmark methods on your data and re-evalute.
* How does your approach affect the latency of the forward pass during inference?
* Why decouple the training in Stage 1 and 2? What happens when Stage 1 training isn’t performed?
* Maybe a naive question, but why do you need quantization at all? Couldn’t one work with continuous embeddings directly? Is this only because you are using pre-trained VQ tokenizers and not doing the training from scratch?
* Minor:
   - Table 3 shows strong results on all categories of the GenEval except for counting? Why is this one so degraded?
   - In L 206, you wrote “ETT achieves superior performance compared to Show-o on wide range of benchmarks”. What I read from Table 2 is that  this only seems to be the case for 2 out of 8 reported benchmarks, and I find this statement too exaggerated.

**Ethical Concerns:**

["NO or VERY MINOR ethics concerns only"]

**Final Justification:**

I thank the author for the thorough response. I raised my rating accordingly.

**Limitations:**

/

**Paper Formatting Concerns:**

/

**Quality:**

3

**Strengths And Weaknesses:**

Strengths:
* Well written paper, easy to follow.
* Strong results (though not SOTA) using less data and fewer LLM parameters compared to the evaluated methods.

Weaknesses:
* It is not clear to me how much of the performance boost reported in the experiments is obtained from the introduced curated SOL-recap data and how much from the proposed method. In other words, the experiments do not decouple data and fine-tuning contributions.
* The novelty is somewhat limited. In my opinion, it is expected that fine tuning the models on downstream tasks improves the results.

---

> ### Author Rebuttal · Authors · 2025-07-31
>
> Thanks for acknowledging our meaningful research perspective, good writing and great efforts for comprehensive evaluation experiments. We have addressed all your questions in detail in the Response section below and have incorporated all feedback in the revision. It is our sincere hope that our thorough explanations to the best of our abilities will contribute to an improved evaluation score.
>
> **Our Responses:**
>
> > [**Q1**]. It is...re-evalute.
>
> [**A1**]. The experiments reported in our paper have been deliberately designed to decouple the benefits brought by data and those by the method. In Tables 4 and 5, we use exactly the same data for fair comparison to demonstrate that our ETT approach indeed brings considerable gains to downstream multimodal understanding and generation tasks by optimizing the VQ’s representations. These gains arise purely from the method rather than the data. Regarding the performance improvements in Tables 2 and 3, it is due to both training data and the ETT method. **It is important to note that many existing large-scale multimodal models use in-house closed-source datasets, and the quality and scale of data vary significantly across methods. From the Data and LLM-Param columns in Tables 2 and 3, it is clear that previous methods use much larger datasets and model sizes compared to ours.** Under these conditions, a strong method must be supported by high-quality data to achieve comparable or better results against prior methods that employ larger LLMs and more data. **The SOL-recap dataset we use is collected from publicly available community datasets, offering no advantage in scale or quality compared to previous SOTA datasets. Moreover, to ensure reproducibility and promote community development, we detail the SOL-recap data collection process in the main paper, which reflects our contribution to the community.** Further, following your suggestion, we present direct performance comparisons with EVE and EMU3 in Table 1 below. **Since we follow their data construction process and collect a comparable amount of data, the comparison is fair and intuitive.** Compared to EMU3, our method achieves better or comparable performance with a smaller LLM size. Compared to EVE, our discrete next-token prediction paradigm attains better multimodal understanding performance with a smaller LLM. We also provide SOTA performance results at Qwen-2.5-3B to demonstrate the strong scaling property of our ETT method.
>
> Table 1: Comparison with SOTA VLMs on various multimodal benchmarks.
> | Method | Category | #LLM-Param | #Und. Data | MMB$^{en}$| SEED$^I$ | MMV | MME | POPE | GQA | SQA | TQA | #Gen. Data | GenEval-Overall |
> |:------|:-------:|:----:|:-----------:|:----:|:-----------:|:----:|:-------:|:----:|:-----------:|:----:|:-----------:|:----:|:----:|
> | EVE  | Continuous |  7B  |  33M / 1.8M  |  52.3  |  64.6  |  25.7  |  1628.0  |  85.0  |  62.6  |  64.9  |  56.8  |  -  |  -  |
> | EMU3 | Discrete |  8B  |  - / -  |  58.5  |  68.2  |  37.2  |  -  |  85.2  |  60.3  |  89.2  |  64.7  |  -  |  0.66  |
> | ETT | Discrete |  1.5B |  32M / 35.3M  |  58.8  |  66.6  |  29.3  |  1532.6  |  82.4  |  59.4  |  91.7  |  56.8  |  30M  |  0.63  |
> | ETT | Discrete |  3B  |  32M / 35.3M  |  61.3  |  67.8  |  32.2  |  1513.9  |  83.2  |  60.5  |  93.3  |  61.1  |  30M  |  0.68  |
>
>
> > [**Q2**]. The novelty...results.
>
> [**A2**]. We respectfully disagree. In fact, this work originates from the key observation that under the next-token prediction (NTP) paradigm using VQ for visual encoding, the main bottleneck in MLLMs lies in the VQ itself, caused by the disjoint training between upstream and downstream stages as well as the current insufficient training paradigm of VQ to learn representations required by downstream tasks. **The insight of improving downstream task performance in the NTP paradigm via end-to-end optimization from the training paradigm perspective is well worth exploring.** This has also been recognized by Reviewer Snt5's positive comment: "I like the concept of end-to-end...the authors address an important gap..." The conventional disjoint training paradigm under NTP is not fundamental and needs to be reformed. **This is a fundamental problem that has been overlooked, under-explored, and is non-trivial in the current research field. Our approach of end-to-end tuning to optimize VQ representations and making the entire scheme practically work is sufficiently innovative both in insight and implementation details. This includes overcoming the challenge of the originally non-differentiable discrete VQ index output and enabling gradient backpropagation to VQ itself, making end-to-end tuning feasible; and maximizing the utilization and optimization of the original visual features of VQ to boost downstream tasks as much as possible. These represent substantial challenges in the current NTP field, all of which are fully addressed in our work. We believe that identifying pain points in the research domain and proposing feasible insights and solutions constitute true innovation and importance. The simplicity and effectiveness of a solution are bonuses, and a straightforward design should not be mistaken for limited technical novelty.**
>
>
> > [**Q3**]. How...inference?
>
> [**A3**]. Our ETT introduces virtually no impact on inference latency for downstream tasks. As shown in Figure 1, the conventional pipeline for MLLMs using VQ involves feeding the indices output by the VQ tokenizer into the LLM, which then performs joint embedding of visual and textual indices followed by sequence modeling. In comparison, the only difference introduced by ETT lies in the embedding strategy: instead of feeding discrete indices, we utilize the original VQ visual codebook embeddings, apply two visual projection layers, and then pass the projected visual embeddings into the LLM for subsequent sequence modeling. **This modification has negligible impact on inference efficiency. Notably, dislike conventional pipeline, our ETT design does not require any modification to the architecture of the original LLM or expanding the visual vocabulary into the textual codebook, which makes ETT more scalable and generalizable.**
>
>
> > [**Q4**]. Why...performed?
>
> [**A4**]. Our current training pipeline follows a standard conventional setting consisting of three stages: Stage 1 for V-L alignment, Stage 2 for learning high-level semantics, and Stage 3 for SFT. This training paradigm is widely adopted in many MLLM works, such as Janus, EVE, Qwen-VL, which further supports that the core insight of our ETT is not tightly constrained by training pipeline. **Removing Stage 1 places a greater burden on subsequent stages to implicitly learn alignment from scratch, resulting in a noticeable degradation in downstream performance especially on multimodal understanding tasks**, where we observe an average drop of 1–2 points.
>
>
> > [**Q5**]. Why...scratch?
>
> [**A5**]. First, continuous embeddings preserve more fine-grained visual features and are intuitively an ideal choice for multimodal understanding tasks. However, in the context of mainstream autoregressive generation, continuous embeddings suffer from a critical issue, error accumulation. As highlighted in works such as [Nexus-gen: A unified model for image understanding, generation, and editing], this problem stems from the training–inference discrepancy. During training, models use ground-truth continuous embeddings as contextual inputs (i.e., teacher forcing), while during inference, they rely on previously generated embeddings. **Small numerical errors in floating-point vectors are recursively propagated and amplified at each generation step, leading to severe degradation in output quality as generation progresses.** In contrast, discrete token index-based next-token prediction effectively mitigates this issue. The discretization step, typically involving a nearest-neighbor lookup, naturally absorbs small prediction errors, thereby preventing their recursive amplification and ensuring stable generation quality. **However, traditional VQ are usually pretrained separately and decoupled from downstream tasks. This means that while discrete token-based models enjoy greater generation stability, their visual token representations may not be well aligned with the semantic needs of complex downstream tasks, resulting in a visual bottleneck that limits performance.**
>
> To address this, we introduce ETT to jointly optimize the vision tokenizer and LLMs in an end-to-end manner. **This breaks the separation between pretrained tokenizers and downstream tasks, allowing discrete tokens to not only retain their robustness against error accumulation but also gain task-adaptive semantic expressiveness. We overcome the non-differentiability of traditional VQ outputs, enabling effective gradient backpropagation and better utilization of VQ's visual features, which significantly enhances the representational capacity of discrete tokens.** In summary, our choice of VQ is not merely due to reliance on a pretrained VQ, but is a deliberate design to circumvent the inevitable error accumulation of continuous embeddings in autoregressive generation and to ensure stable generation quality. **The innovation of our work lies in enhancing and adapting the discrete token paradigm to better serve both multimodal understanding and generation, an aspect that is both necessary and valuable in current research community.**
>
>
> > [**Q6**]. Table 3...exaggerated.
>
> [**A6**]. 1. The current suboptimal performance on the counting category is primarily due to the limited availability of relevant training data. We believe that incorporating more text-image data specifically designed for counting patterns would lead to significant performance improvements in this category. 2. To ensure the rigor of our claims, we have removed the phrase "across a wide range of benchmarks" from the main text.

---

> > ### Author Response · Authors · 2025-08-01
> > **Revisiting Reviewer t4q5‘s Comments in Detail**
> >
> > Due to the character limit during the rebuttal phase, we had to minimize the enumeration of the questions you raised. To make it easier for you to check our point-by-point responses, we have provided a detailed list of your original comments below. We have addressed all your questions in detail in the Response section above and have incorporated all feedback in the revision. It is our sincere hope that our thorough explanations to the best of our abilities will contribute to an improved evaluation score.
> >
> > > [**Q1**]. It is not clear to me how much of the performance boost reported in the experiments is obtained from the introduced curated SOL-recap data and how much from the proposed method. In other words, the experiments do not decouple data and fine-tuning contributions. As I wrote above, it is hard to judge from where the improvements in Table 2 and 3 come from. Is it because your curated data is of higher quality or is it because of the ETT method or both? I think it would be very valuable to decouple these, for example with an experiment where you train one of the benchmark methods on your data and re-evalute.
> >
> > > [**Q2**]. The novelty is somewhat limited. In my opinion, it is expected that fine tuning the models on downstream tasks improves the results.
> >
> > > [**Q3**]. How does your approach affect the latency of the forward pass during inference?
> >
> > > [**Q4**]. Why decouple the training in Stage 1 and 2? What happens when Stage 1 training isn’t performed?
> >
> > > [**Q5**]. Maybe a naive question, but why do you need quantization at all? Couldn’t one work with continuous embeddings directly? Is this only because you are using pre-trained VQ tokenizers and not doing the training from scratch?
> >
> > > [**Q6**]. Minor: 1. Table 3 shows strong results on all categories of the GenEval except for counting? Why is this one so degraded? 2. In L 206, you wrote “ETT achieves superior performance compared to Show-o on wide range of benchmarks”. What I read from Table 2 is that this only seems to be the case for 2 out of 8 reported benchmarks, and I find this statement too exaggerated.

---

> > > ### Author Response · Authors · 2025-08-08
> > > **Sincere Request for Review of Our Wholehearted Detailed Response**
> > >
> > > Dear Reviewer t4q5,
> > >
> > > We would like to touch base with you to see whether you had a chance to look at our wholehearted response. We hope that it has helped address all the concerns you have raised in your reviews. If there are other concerns or if you have more questions, we will be more than happy to provide additional clarification.
> > >
> > > Thanks again for your valuable time! We sincerely hope that you will find our work deserving of your esteemed recognition and that it may receive a promising final rating.
> > >
> > > Best,
> > >
> > > Authors

---

### Official Review · Reviewer_xzNc · 2025-07-03

**Clarity:** 3
**Significance:** 2
**Originality:** 2
**Rating:** 5
**Confidence:** 4

**Summary:**

This paper introduces ETT, an end-to-end vision tokenizer that jointly optimizes both image reconstruction and downstream auto-regressive tasks. While prior works typically rely on pre-trained vision tokenizers that are kept frozen during downstream training, this paper demonstrates that integrating the tokenizer into the end-to-end training loop is straightforward and beneficial. By doing so, ETT helps unlock the tokenizer's potential for downstream understanding tasks, which are often hindered when the tokenizer is only trained for reconstruction objectives. Experiments show that ETT achieves competitive performance in both image understanding and generation.

**Questions:**

Please see weaknesses.

**Ethical Concerns:**

["NO or VERY MINOR ethics concerns only"]

**Final Justification:**

This paper presents an intuitive approach that addresses an important area currently lacking sufficient research. Thorough experiments are conducted to validate both the effectiveness and extensibility of the proposed method. Therefore, I recommend Accept.

**Limitations:**

Please see weaknesses.

**Paper Formatting Concerns:**

No major paper formatting concerns are noted.

**Quality:**

3

**Strengths And Weaknesses:**

## Strengths

- The paper is well-organized and easy to follow, with a clean methodology, straightforward solutions, and clear experimental setups and comparisons.

## Weaknesses

- The technical novelty is somewhat limited. While the paper demonstrates that end-to-end training is feasible and beneficial, the core idea is relatively incremental.
- In Table 4, the performance drop from row 1 to row 2—when replacing visual tokens from indices to embeddings—is surprisingly large. The paper should better explain why this change has such a significant impact on generation performance.
- In the ablation study exploring the trade-off between understanding and generation, only rFID is reported for generation quality. It would be more informative to also include sample image outputs or additional quantitative metrics (e.g., FID, IS) to illustrate how generation quality degrades.
- The paper should discuss how ETT could integrate with recent trends in jointly trained tokenizers for both understanding and generation, such as VILA-U or UniTok. Would the proposed method still provide gains in such setups?

---

> ### Author Rebuttal · Authors · 2025-07-31
>
> Thanks very much for acknowledging our nice research perspective, good writing and great efforts for comprehensive evaluation experiments. We have addressed all your questions in detail in the Response section below and have incorporated all feedback in the revision. It is our sincere hope that our thorough explanations to the best of our abilities will contribute to an improved evaluation score of our work.
>
>
> **Our Responses to Paper Weaknesses:**
>
> > [**Q1**]. The technical novelty is somewhat limited. While the paper demonstrates that end-to-end training is feasible and beneficial, the core idea is relatively incremental.
>
> [**A1**]. We respectfully disagree. In fact, this work originates from the key observation that under the next-token prediction (NTP) paradigm using discrete vision tokenizers for visual encoding, the main bottleneck in MLLMs lies in the discrete vision tokenizer itself, caused by the disjoint training between upstream and downstream stages as well as the current insufficient training paradigm of VQ to learn representations required by downstream tasks. The insight of improving downstream task performance in the NTP paradigm via end-to-end optimization from the training paradigm perspective is well worth exploring. This has also been recognized by Reviewer Snt5's positive comment: "**I like the concept of end-to-end optimization of vision tokenizers, which is a significant advancement in the field. By integrating tokenizer training with downstream tasks, the authors address an important gap in existing methodologies.**" As we emphasized in the Introduction, the conventional disjoint training paradigm under NTP is not fundamental and needs to be reformed. **This is a fundamental problem that has been overlooked, under-explored, and is non-trivial in the current research field. Our approach of end-to-end tuning to optimize discrete vision tokenizer representations and making the entire scheme practically work is sufficiently innovative both in insight and implementation details. This includes overcoming the challenge of the originally non-differentiable discrete VQ index output and enabling gradient backpropagation to VQ itself, making end-to-end tuning feasible; and maximizing the utilization and optimization of the original visual features of VQ to boost downstream tasks as much as possible. These represent substantial challenges in the current NTP field, all of which are fully addressed in our work. We believe that identifying pain points in the research domain and proposing feasible insights and solutions constitute true innovation and importance. The simplicity and effectiveness of a solution are bonuses, and a straightforward design should not be mistaken for limited technical novelty.**
>
>
> > [**Q2**]. In Table 4, the performance drop from row 1 to row 2—when replacing visual tokens from indices to embeddings—is surprisingly large. The paper should better explain why this change has such a significant impact on generation performance.
>
> [**A2**]. We appreciate the reviewer’s insightful comment. The sharp performance drop from row 1 to row 2 in Table 4 is primarily due to the fact that autoregressive generation requires precise modeling of low-level visual details, which are better preserved through discrete token indices. **In IBQ, the indices are directly optimized via gradient-based index supervision, which enforces strong token separability and facilitates the LLM’s ability to model fine-grained token transitions during generation.** In contrast, when switching to raw embeddings from a non-tuned IBQ tokenizer, **such embeddings lack task-specific discriminative structure and are not explicitly optimized for generation, making it harder for the model to capture visual token dynamics effectively**. However, **once we apply ETT tuning, the embeddings are adapted jointly with the generation objective, allowing generation-aware signals to be injected back into the tokenizer and improving the quality of the embeddings accordingly**. This alignment enables the embeddings to better approximate the role of discrete indices, thus recovering generation performance despite the shift in representation form.
>
>
> > [**Q3**]. In the ablation study exploring the trade-off between understanding and generation, only rFID is reported for generation quality. It would be more informative to also include sample image outputs or additional quantitative metrics (e.g., FID, IS) to illustrate how generation quality degrades.
>
> [**A3**]. We believe the phrase "ablation study exploring the trade-off between understanding and generation" should be referring to "ablation study exploring the trade-off between multimodal understanding and visual reconstruction". Regarding the ablation experiment in Table 5, we have already presented sample image outputs corresponding to Table 5's first and third rows of the numerical results in Figure 3. **Based on your valuable suggestion, we further supplement Table 5 with additional commonly used reconstruction metrics, including SSIM and PSNR, and add visual samples corresponding to each row of Table 5’s results in Figure 3**, making the ablation study more informative and convincing. Since we cannot upload images or external links during this rebuttal phase, we have included the quantitative results in the table below. The revised version will incorporate the supplemented quantitative metrics and more informative visualizations.
>
> Table 1: Ablation study on the impact of ETT for the trade-offs between multimodal perception and visual reconstruction.
> | Tuning Tasks for VQ | α | SEED$^I$ | GQA | TQA | MME$^P$ | ImageNet-rFID↓ | ImageNet-PSNR↑ | ImageNet-SSIM↑ |
> |:------|:-------:|:----:|:-----------:|:----:|:-----------:|:----:|:----:|:----:|
> |  -  | - |  54.8  |  52.6  |  40.9  |  1034.0  |  1.03  |  21.73 |  0.60  |
> | Und. only | - |  61.2  |  55.2  |  48.0  |  1164.3  |  45.70  |  13.22  |  0.19  |
> | Und.&Rec. | 0.25 |  60.0  |  54.8  |  46.9  |  1124.7  |  1.65  |  20.89  |  0.58  |
> | Und.&Rec. | 0.5 |  59.9  |  54.7  |  46.9  |  1118.1  |  1.66  |  21.14  |  0.59  |
> | Und.&Rec. | 1.0 |  59.3  |  53.9  |  45.7  |  1088.2  |  1.50  |  21.48  |  0.60  |
>
>
> > [**Q4**]. The paper should discuss how ETT could integrate with recent trends in jointly trained tokenizers for both understanding and generation, such as VILA-U or UniTok. Would the proposed method still provide gains in such setups?
>
> [**A4**]. Thank you for your valuable suggestion. Our ETT framework can be combined with recent trends of jointly trained tokenizers for both understanding and generation and bring performance improvements. **Actually, these two approaches are orthogonal and do not interfere with each other.** The recent jointly trained tokenizers learn high-level semantics by distilling CLIP vision-language alignment during the VQ pretraining stage, which is separated from downstream training. Although this enables VQ to learn beyond mere reconstruction, the disjoint upstream-downstream training paradigm still prevents VQ from fully acquiring the comprehensive representations required downstream, thus somewhat creating a visual bottleneck for downstream tasks. Therefore, our proposed end-to-end tuning that directly optimizes the vision tokenizer’s representations is highly meaningful. Since architectures like VILA-U and UniTok introduce additional uncertainties when combined with ETT (e.g., requiring extra transformer blocks), to quickly validate during the rebuttal phase that our ETT framework is general and promising, we have additionally conducted experiments using another one of the jointly trained tokenizers for both understanding and generation as the discrete vision tokenizer candidate. These results are currently on the way and will be updated during the discussion phase for your reference. **Although the jointly trained tokenizers can learn more comprehensive and holistic visual representations during pretraining, our ETT can still bring non-negligible downstream task performance gains.** We will include the corresponding analyses and quantitative results in the revised version.

---

> > ### Comment · Reviewer_xzNc · 2025-08-04
> >
> > I thank the authors for their thorough response. My concerns have been mostly addressed, with the exception of the pending experimental results on the joint tokenizer. I fully agree that the two approaches are orthogonal and do not interfere with each other—I would simply like to see the performance gain demonstrated empirically. I will raise my score and recommend that the authors include all additional experiments shared in the rebuttal (including those provided in response to other reviews) in the revised version, as they significantly strengthen the paper.

---

> ### Author Response · Authors · 2025-08-04
> **Sincere Gratitude for Your Generous Recognition and Substantial Rating Improvement of Our Work**
>
> **Thank you very much for raising your final score and recommending acceptance with the valuable comment "My concerns have been mostly addressed".** We sincerely appreciate your great recognition of our work and the efforts we have dedicated during the invaluable rebuttal period. **We are truly gratified that our revised manuscript has addressed most of your concerns and convinced you of the solidity of our paper's content.**
>
> We will release the experimental results that are being supplemented on the way as soon as possible during the discussion period and add them all to the final revised version. **We expect to promptly report the results once we obtain the understanding performance gains brought by our ETT on jointly trained tokenizers for both understanding and generation, as well as the experimental results from replacing the current Qwen2.5 LLM with other LLM candidates in ETT, so that reviewers can conveniently reference them. Due to the very tight timeline during the discussion period, we have been actively supplementing a number of additional experiments and kindly ask for your patience.**
>
> If you have any further questions, please do not hesitate to let us know and we will spare no effort to respond promptly.

---

> > ### Author Response · Authors · 2025-08-08
> > **Additional Experiments with Different VQ Candidates**
> >
> > Due to the extremely limited time during the discussion period, to demonstrate that ETT is both scalable and model-agnostic, we have promptly supplemented the ablation studies using UniTok as the VQ candidate and understanding-only end-to-end tuning on 12M multimodal understanding data. **Together with the previously reported ablation results using Qwen2.5-3B and Vicuna-7B-v1.5 as LLM candidates on the same data, these findings further confirm the strong potential of our proposed ETT approach. It is worth emphasizing that, to obtain results as soon as possible, we follow the same training settings used for IBQ when conducting experiments with UniTok as VQ candidate. However, we believe that under the ETT architecture, the optimal training settings should vary across different VQs. Therefore, the actual upper bound of performance gains that ETT can achieve with different VQs could be even higher.** At this point, we have relentlessly completed all the additional experiments we promised to provide during the rebuttal period. It is our sincere hope that our thorough explanations and added experiment results to the best of our abilities will contribute to an improved evaluation score.
> >
> > **Table 1: Ablation study on the benefits of ETT for multimodal understanding with different discrete vision tokenizer candidates.**
> > | ETT | VQ | LLM | SEED$^I$ | GQA | TQA | MME$^P$ |
> > |:------|:-----------:|:-----------:|:----:|:-----------:|:----:|:-----------:|
> > |   | UniTok | Qwen2.5-1.5B |  66.7  |  57.5  |  50.6  |  1274.5  |
> > | √ | UniTok | Qwen2.5-1.5B |  69.8  |  60.5  |  57.9  |  1327.7  |

---

### Official Review · Reviewer_Snt5 · 2025-07-03

**Clarity:** 3
**Significance:** 3
**Originality:** 3
**Rating:** 5
**Confidence:** 5

**Summary:**

This paper presents an innovative methodology, termed End-to-End Vision Tokenizer Tuning (ETT), designed to address the inherent limitations of conventional vision tokenization techniques. The authors contend that the prevalent practice of optimizing tokenizers independently of downstream applications can result in substantial misalignments. Such discrepancies are particularly detrimental in tasks requiring sophisticated visual representations, including image generation and visual question answering. ETT facilitates the joint optimization of both the vision tokenizer and downstream autoregressive tasks, thereby leveraging visual embeddings to improve multimodal understanding and generative capabilities. The authors provide empirical evidence demonstrating a 2-6% performance enhancement over traditional frozen tokenizer pipelines, without compromising the original reconstruction quality.

**Questions:**

None

**Ethical Concerns:**

["NO or VERY MINOR ethics concerns only"]

**Final Justification:**

Authors made a good rebuttal, where all my concerns have been solved properly. I thus give ACCEPT as my final recommendation.

**Limitations:**

yes

**Quality:**

3

**Strengths And Weaknesses:**

PROS:

1. I like the concept of end-to-end optimization of vision tokenizers, which is a significant advancement in the field. By integrating tokenizer training with downstream tasks, the authors address an important gap in existing methodologies.

2. The extensive experiments show substantial improvements in multimodal understanding and generation tasks, with metrics indicating a clear advantage over existing models. This suggests that ETT can effectively enhance model performance with relatively simple modifications.

3. The authors provide extensive experimental results across multiple benchmark datasets, reinforcing the robustness of their findings and demonstrating the versatility of their approach.

4. ETT maintains the original reconstruction performance of the vision tokenizer, which is essential for tasks that rely on high-quality visual outputs.

5. The paper is well-written, which makes it easy to follow.

- CONS:

1. The experiments are conducted using a specific vision tokenizer (IBQ-based) and a specific LLM (Qwen2.5-1.5B). While the results are strong, the paper does not explore whether the ETT method is equally effective with other popular vision tokenizers (e.g., VQGAN, MaskGIT) or different LLM families. While this is a reasonable scope limitation for a single paper, it leaves the question of the method's generalizability open. The authors rightly point to this as a direction for future work.

2. The paper provides good qualitative examples of generated images and reconstruction quality. However, the analysis could be more impactful. For instance, to support the claim that ETT improves high-level semantics like text rendering, a direct side-by-side comparison showing a failure case of the baseline model (e.g., garbled text in an image) that is corrected by the ETT-tuned model would be highly compelling.

3. The paper presents a promising proof of concept for ETT, but questions persist regarding its scalability and generalizability. Key concerns include how ETT’s benefits evolve with larger LLMs (e.g., 70B+ parameters) and bigger datasets, as a powerful LLM might mitigate a frozen tokenizer’s limitations. Additionally, its applicability to non-autoregressive architectures (like MaskGIT) and other modalities (video, audio) remains unclear, with challenges such as capturing temporal consistency in video. At least, they should be discussed.

---

> ### Author Rebuttal · Authors · 2025-07-31
>
> We sincerely appreciate your friendliness and recognition of our meaningful research perspective, good writing, great potential of our ETT and great efforts for comprehensive evaluation experiments. We have addressed all of your questions in detail in the following Response section and have incorporated all feedback in the revision. We hope that our detailed explanations will give us the precious opportunity to raise the evaluation score of our work in your perspective.
>
> **Our Responses to Paper Weaknesses:**
>
> > [**Q1**]. The experiments are conducted using a specific vision tokenizer (IBQ-based) and a specific LLM (Qwen2.5-1.5B). While the results are strong, the paper does not explore whether the ETT method is equally effective with other popular vision tokenizers (e.g., VQGAN, MaskGIT) or different LLM families. While this is a reasonable scope limitation for a single paper, it leaves the question of the method's generalizability open. The authors rightly point to this as a direction for future work.
>
> [**A1**]. Many thanks for raising the important question of generalizability, and for your kindness toward this exploratory work. As we claimed in the main paper, since we innovatively propose to leverage the original VQ's visual codebook embeddings, rather than the discrete indices that make end-to-end tuning impossible, our ETT requires appropriate selection of discrete vision tokenizers. For instance, **a larger codebook embedding dimension allows ETT to better unleash its potential**. However, current VQ models that meet this criterion are extremely limited, as most previous efforts have focused on designing better VQs under the framework of using discrete indices, without paying attention to scaling up the vision codebook embedding dimension. Since our work mainly aims to enhance performance on downstream multimodal understanding and generation tasks under the next-token prediction paradigm by optimizing both the training paradigm and the way VQ features are utilized, we adopt IBQ, one of the more suitable candidates, as our discrete vision tokenizer. However, **we would like to emphasize that our ETT is also capable of bringing downstream performance gains to other VQ designs**. To further demonstrate that our ETT architecture is generalizable and promising, we have additionally conducted experiments using one of the jointly trained tokenizers for both understanding and generation as the discrete vision tokenizer candidate. These results are currently on the way and will be updated during the discussion phase for your reference. Moreover, our ETT architecture imposes no specific constraints on the choice of LLM. Based on your valuable suggestion, we have included ablation results using Qwen2.5-3B and Vicuna-7B-v1.5 as LLM candidates on 12M-scale multimodal generation and understanding data (Table 1), as well as SOTA performance using Qwen2.5-3B on the full dataset (Table 2) to further demonstrate that our ETT method is both scalable and model-agnostic. The results for Qwen2.5-3B are shown below, while those for Vicuna-7B-v1.5 are also on the way and will be updated during the discussion phase. All corresponding results will be added to the final revised version. **As an exploratory effort to optimize the training paradigm of visual tokenizers under the discrete next-token prediction framework, there remain many opportunities for future exploration, and we are committed to further advancing the capabilities of ETT.**
>
>
> Table 1: Ablation study on the benefits of ETT for multimodal understanding and generation with different large language models.
> | ETT | LLM | SEED$^I$ | GQA | TQA | MME$^P$ | GenEval-Overall |
> |:------|:-------:|:----:|:-----------:|:----:|:-----------:|:----:|
> |   | Qwen2.5-3B |  53.9  |  52.4  |  40.6  |  986.9  |  0.29  |
> | √ | Qwen2.5-3B |  64.0  |  56.7  |  49.4  |  1188.9  |  0.32  |
>
>
> Table 2: Comparison with existing state-of-the-art vision-language models on various multimodal benchmarks.
> | Method | Category | #LLM-Param | #Und. Data | MMB$^{en}$| SEED$^I$ | MMV | MME | POPE | GQA | SQA | TQA | #Gen. Data | GenEval-Overall |
> |:------|:-------:|:----:|:-----------:|:----:|:-----------:|:----:|:-------:|:----:|:-----------:|:----:|:-----------:|:----:|:----:|
> | EVE  | Continuous VLMs |  7B  |  33M / 1.8M  |  52.3  |  64.6  |  25.7  |  1628.0  |  85.0  |  62.6  |  64.9  |  56.8  |  -  |  -  |
> | Janus | Continuous VLMs |  1.3B  |  - / -  |  69.4  |  63.7  |  34.3  |  1338.0  |  87.0  |  59.1  |  -  |  -  |  -  |  0.61  |
> | EMU3 | Discrete VLMs |  8B  |  - / -  |  58.5  |  68.2  |  37.2  |  -  |  85.2  |  60.3  |  89.2  |  64.7  |  -  |  0.66  |
> | ETT (ours) | Discrete VLMs |  1.5B |  32M / 35.3M  |  58.8  |  66.6  |  29.3  |  1532.6  |  82.4  |  59.4  |  91.7  |  56.8  |  30M  |  0.63  |
> | ETT (ours) | Discrete VLMs |  3B  |  32M / 35.3M  |  61.3  |  67.8  |  32.2  |  1513.9  |  83.2  |  60.5  |  93.3  |  61.1  |  30M  |  0.68  |
>
>
> > [**Q2**]. The paper provides good qualitative examples of generated images and reconstruction quality. However, the analysis could be more impactful. For instance, to support the claim that ETT improves high-level semantics like text rendering, a direct side-by-side comparison showing a failure case of the baseline model (e.g., garbled text in an image) that is corrected by the ETT-tuned model would be highly compelling.
>
> [**A2**]. Many thanks for your valuable suggestion. Per your advice, we have further included a side-by-side visual comparison showing a typical failure case of the baseline model in rendering text, which is effectively corrected by the ETT-tuned model. This visual evidence supports our claim that ETT enhances high-level semantic fidelity such as accurate text rendering. As we are unable to include external links during the rebuttal phase, we have incorporated the corresponding visualizations into the revised version, hoping that this analysis could be more impactful.
>
>
> > [**Q3**]. The paper presents a promising proof of concept for ETT, but questions persist regarding its scalability and generalizability. Key concerns include how ETT’s benefits evolve with larger LLMs (e.g., 70B+ parameters) and bigger datasets, as a powerful LLM might mitigate a frozen tokenizer’s limitations. Additionally, its applicability to non-autoregressive architectures (like MaskGIT) and other modalities (video, audio) remains unclear, with challenges such as capturing temporal consistency in video. At least, they should be discussed.
>
> [**A3**]. Many thanks for your valuable comment. First, regarding how ETT’s benefits evolve with larger LLMs (e.g., 70B+ parameters) and bigger datasets, we believe that with larger LLM sizes, our ETT can still bring improvements to downstream multimodal understanding and generation by optimizing the visual representation of discrete vision tokenizers themselves. Although, as you pointed out, a powerful LLM might mitigate the limitations of a frozen tokenizer, our focus is to explore how to maximize the potential of discrete vision tokenizers under the next-token prediction paradigm within relatively limited resource settings. While the community offers many LLMs of various sizes, even the largest current LLMs still heavily rely on strong visual capabilities to perform downstream multimodal tasks effectively. Therefore, **the benefits of ETT may be somewhat diminished with extremely large LLMs, but they remain significant and cannot be ignored.** Experimental results with 1.5B LLMs in the main text and additional experiments with 3B and 7B models added during the rebuttal phase sufficiently demonstrate the strong scalability and generalizability of our ETT framework. Notably, **optimizing the vision tokenizer representation based on a smaller LLM (e.g., 3B or 7B) to boost the visual capabilities of VQ for larger LLMs (e.g., 14B, 32B) is a promising future direction we plan to explore, which could greatly improve training efficiency and reduce costs.** Moreover, **with increased data scale, the adopted discrete vision tokenizer continuously learns more generalized vision-language alignment semantics and a better balance of high- and low-level representations through the LLM-provided caption loss, further unleashing ETT’s potential**. We have indeed observed such data scaling properties where ETT’s gains improve as data size grows. Finally, regarding ETT’s applicability to non-autoregressive architectures and other modalities, we believe our ETT framework can generalize to other architectures and modalities. Although our current focus is on enhancing the semantic awareness of discrete vision tokenizers via end-to-end tuning with image-to-text caption loss, **the supervision form and included modalities are not restricted.** For example, ETT can inject necessary representations for downstream tasks into non-autoregressive architectures (like MaskGIT), and it can also be extended to discrete vision tokenizers for video or audio modalities to provide the high-level semantic representations required for multimodal understanding and generation. We will incorporate these analyses into the revised version following your suggestions.

---

> ### Author Response · Authors · 2025-08-06
> **Additional Experiments with Different LLM Candidates**
>
> Due to the extremely limited time during the discussion period, to demonstrate that ETT is both scalable and model-agnostic, we have promptly supplemented the ablation studies using Vicuna-7B-v1.5 as the LLM candidate on 12M vision-language understanding data. **It is worth emphasizing that, to obtain results as soon as possible, we follow the same training settings used for Qwen2.5-1.5B when conducting experiments with Qwen2.5-3B and Vicuna-7B-v1.5 as LLM candidates. However, we believe that under the ETT architecture, the optimal training settings should vary across different LLMs. Therefore, the actual upper bound of performance gains that ETT can achieve with different LLMs could be even higher.** **Together with the previously reported ablation results using Qwen2.5-3B on the same data, these findings further confirm the strong potential of our proposed ETT approach.** As promised, the remaining experiment--taking one of the jointly trained tokenizers for both understanding and generation as the discrete visual tokenizer candidate--is still on the way. **In the remaining short time, we will prioritize reporting the corresponding performance gains brought by ETT on multimodal understanding tasks.**
>
> **Table 1: Ablation study on the benefits of ETT for multimodal understanding with different LLMs.**
> | ETT | LLM | SEED$^I$ | GQA | TQA | MME$^P$ |
> |:------|:-------:|:----:|:-----------:|:----:|:-----------:|
> |   | Vicuna-7B-v1.5 |  45.3  |  46.2  |  40.7  |  828.5 |
> | √ | Vicuna-7B-v1.5 |  49.5  |  49.5  |  41.5  |  904.0 |

---

> > ### Author Response · Authors · 2025-08-08
> > **Additional Experiments with Different VQ Candidates**
> >
> > Due to the extremely limited time during the discussion period, to demonstrate that ETT is both scalable and model-agnostic, we have promptly supplemented the ablation studies using UniTok as the VQ candidate and understanding-only end-to-end tuning on 12M multimodal understanding data. **Together with the previously reported ablation results using Qwen2.5-3B and Vicuna-7B-v1.5 as LLM candidates on the same data, these findings further confirm the strong potential of our proposed ETT approach. It is worth emphasizing that, to obtain results as soon as possible, we follow the same training settings used for IBQ when conducting experiments with UniTok as VQ candidate. However, we believe that under the ETT architecture, the optimal training settings should vary across different VQs. Therefore, the actual upper bound of performance gains that ETT can achieve with different VQs could be even higher.** At this point, we have relentlessly completed all the additional experiments we promised to provide during the rebuttal period. It is our sincere hope that our thorough explanations and added experiment results to the best of our abilities will contribute to an improved evaluation score.
> >
> > **Table 1: Ablation study on the benefits of ETT for multimodal understanding with different discrete vision tokenizer candidates.**
> > | ETT | VQ | LLM | SEED$^I$ | GQA | TQA | MME$^P$ |
> > |:------|:-----------:|:-----------:|:----:|:-----------:|:----:|:-----------:|
> > |   | UniTok | Qwen2.5-1.5B |  66.7  |  57.5  |  50.6  |  1274.5  |
> > | √ | UniTok | Qwen2.5-1.5B |  69.8  |  60.5  |  57.9  |  1327.7  |

---

### Official Review · Reviewer_Wh8j · 2025-07-08

**Clarity:** 3
**Significance:** 2
**Originality:** 2
**Rating:** 5
**Confidence:** 3

**Summary:**

This paper presents a new approach to visual tokenization that can be applied to both visual generation and understanding. Instead of using discrete indices from a VQ tokenizer, the method proposes using codebook embeddings along with a projector layer. The approach shows improvements on various visual understanding and generation benchmarks.

**Questions:**

1. Why isn’t there any comparison with other types of visual tokenization methods, such as continuous embeddings used in popular models like Qwen, LLaVA, or Gemma? I understand that there are limited works that handle both generation and understanding, but there should at least be a comparison with methods that address one of these tasks.
2. Have you observed any synergy from jointly tuning for both reconstruction and understanding? What benefits does this type of joint tuning offer compared to tuning for a single task?

**Ethical Concerns:**

["NO or VERY MINOR ethics concerns only"]

**Final Justification:**

I thank the authors for their thorough response. While I remain unconvinced by the rationale for merging visual understanding and generation into a single framework, I acknowledge that it is an established problem in the field, and the authors' work represents a meaningful advancement in this area. I strongly recommend that the revised paper include results using a jointly trained tokenizer for both understanding and generation as the discrete visual tokenizer candidate, as this would further strengthen the contribution. I am raising my score to 5 and recommend acceptance.

**Limitations:**

Yes

**Paper Formatting Concerns:**

TeX is supported

**Quality:**

3

**Strengths And Weaknesses:**

Strengths:
The paper addresses a relevant gap in autoregressive multimodal models by proposing to align vision tokenization with downstream tasks, rather than relying on independently trained tokenizers. This is a practical consideration for models performing both visual understanding and generation.
The method is straightforward, involving the use of codebook embeddings and a projector layer to interface with a decoder-only language model. It does not require architectural changes to the language model.
The approach shows consistent improvements over frozen tokenizer baselines across multiple benchmarks. These gains suggest that end-to-end tuning of the tokenizer can improve performance without sacrificing too much understanding quality.


Weaknesses:
Limited Generalizability: The method is only evaluated with a single LLM (Qwen2.5-1.5B) within a large-scale, tightly controlled training pipeline. It remains unclear whether ETT is truly model-agnostic or if its effectiveness is contingent on the specific architecture and training setup used. The lack of validation across different language models raises concerns about the robustness and transferability of the approach.
No Comparison with Continuous Visual Encoders: The paper does not include comparisons with CLIP-style vision encoders or other continuous visual representation methods, which are widely adopted in models such as LLaVA, or Qwen VL. Without such baselines, it's difficult to assess the true benefit of discrete tokenization and end-to-end tuning proposed in ETT, especially when simpler or more modular alternatives exist.
Questionable Need for Joint Objectives: Table 5 suggests limited synergy between visual reconstruction and multimodal understanding. In fact, optimizing for one often comes at the expense of the other. This raises the question of whether it is necessary, or even desirable, to jointly optimize for both tasks within a single model. The paper does not justify why a unified approach is preferable over specialized models optimized independently for understanding or generation.

---

> ### Author Rebuttal · Authors · 2025-07-31
>
> We sincerely appreciate your friendliness and recognition of our meaningful research perspective, great potential of our ETT and the comprehensive experiments. We have addressed all of your questions in detail in the following Response section and have incorporated all feedback in the revision. We genuinely hope that our detailed explanations and the additionally supplemented results with our best efforts will give us the precious opportunity to raise the evaluation score of our work.
>
> **Our Responses to Paper Weaknesses:**
>
> > [**Q1**]. Limited Generalizability: The method is only evaluated with a single LLM within a large-scale, tightly controlled training pipeline. It remains unclear whether ETT is truly model-agnostic or if its effectiveness is contingent on the specific architecture and training setup used. The lack of validation across different language models raises concerns about the robustness and transferability of the approach.
>
> [**A1**]. First, we would like to clarify that describing our setup as a “large-scale, tightly controlled training pipeline” is not fair. Our training pipeline follows a standard and conventional three-stage setting widely adopted in both continuous and discrete MLLM works, including [1]–[3]: stage 1 focuses on vision-language alignment, stage 2 on high-level semantic learning, and stage 3 on supervised fine-tuning. **Therefore, the core insight of our proposed end-to-end tuning is not tightly constrained by the design of the training pipeline.** Second, our ETT method does not impose any requirements on the choice of LLM. In response to your valuable suggestion, **we have added ablation studies using both Qwen2.5-3B and Vicuna-7B-v1.5 as LLM candidates on 12M vision-language understanding and generation data (Table 1), as well as SOTA results using Qwen2.5-3B on the full dataset (Table 2), to demonstrate that ETT is scalable and model-agnostic**. The results for Qwen2.5-3B have been presented as follows, and those for Vicuna-7B-v1.5 are currently in progress and will be updated during discussion phase. Additionally, **to further verify the generalizability of our ETT framework, we are also conducting experiments using one of the jointly trained tokenizers for both understanding and generation as the discrete visual tokenizer candidate. These results are also on the way and will be made available during discussion.** All the results will be included in the revised version.
>
>
> Table 1: Ablation study on the benefits of ETT for multimodal understanding and generation with different LLMs.
> | ETT | LLM | SEED$^I$ | GQA | TQA | MME$^P$ | GenEval-Overall |
> |:------|:-------:|:----:|:-----------:|:----:|:-----------:|:----:|
> |   | Qwen2.5-3B |  53.9  |  52.4  |  40.6  |  986.9  |  0.29  |
> | √ | Qwen2.5-3B |  64.0  |  56.7  |  49.4  |  1188.9  |  0.32  |
>
>
> Table 2: Comparison with existing state-of-the-art VLMs on various multimodal benchmarks.
> | Method | Category | #LLM-Param | #Und. Data | MMB$^{en}$| SEED$^I$ | MMV | MME | POPE | GQA | SQA | TQA | #Gen. Data | GenEval-Overall |
> |:------|:-------:|:----:|:-----------:|:----:|:-----------:|:----:|:-------:|:----:|:-----------:|:----:|:-----------:|:----:|:----:|
> | EVE  | Continuous VLMs |  7B  |  33M / 1.8M  |  52.3  |  64.6  |  25.7  |  1628.0  |  85.0  |  62.6  |  64.9  |  56.8  |  -  |  -  |
> | Janus | Continuous VLMs |  1.3B  |  - / -  |  69.4  |  63.7  |  34.3  |  1338.0  |  87.0  |  59.1  |  -  |  -  |  -  |  0.61  |
> | EMU3 | Discrete VLMs |  8B  |  - / -  |  58.5  |  68.2  |  37.2  |  -  |  85.2  |  60.3  |  89.2  |  64.7  |  -  |  0.66  |
> | ETT (ours) | Discrete VLMs |  1.5B |  32M / 35.3M  |  58.8  |  66.6  |  29.3  |  1532.6  |  82.4  |  59.4  |  91.7  |  56.8  |  30M  |  0.63  |
> | ETT (ours) | Discrete VLMs |  3B  |  32M / 35.3M  |  61.3  |  67.8  |  32.2  |  1513.9  |  83.2  |  60.5  |  93.3  |  61.1  |  30M  |  0.68  |
>
>
> [1] Wu C, Chen X, Wu Z, et al. Janus: Decoupling visual encoding for unified multimodal understanding and generation[C]//Proceedings of the Computer Vision and Pattern Recognition Conference. 2025: 12966-12977.
>
> [2] Diao H, Cui Y, Li X, et al. Unveiling encoder-free vision-language models[J]. Advances in Neural Information Processing Systems, 2024, 37: 52545-52567.
>
> [3] Bai J, Bai S, Chu Y, et al. Qwen technical report[J]. arXiv preprint arXiv:2309.16609, 2023.
>
>
> > [**Q2**]. No Comparison with Continuous Visual Encoders: The paper does not include comparisons with CLIP-style vision encoders or other continuous visual representation methods, which are widely adopted in models such as LLaVA, or Qwen VL. Without such baselines, it's difficult to assess the true benefit of discrete tokenization and end-to-end tuning proposed in ETT, especially when simpler or more modular alternatives exist.
>
> [**A2**]. Thanks for this comment. Indeed, continuous embeddings preserve more fine-grained visual features, making them an intuitive and ideal choice for multimodal understanding tasks. However, in the context of mainstream autoregressive generation tasks, continuous embeddings suffer from a critical issue of error accumulation. Prior work such as [Nexus-gen: A unified model for image understanding, generation, and editing] has extensively discussed this problem: during training, models typically use ground-truth continuous embeddings as context inputs via teacher forcing, whereas during inference, the model must rely on its own predicted embeddings for the next-step input. Small numerical discrepancies in floating-point vectors at each generation step can recursively propagate through the context, and such errors gradually accumulate, ultimately leading to severe degradation in image generation quality. In contrast, the next-token prediction (NTP) paradigm using discrete token indices is much more resilient to this issue. The process of retrieving the nearest neighbor token index at each step effectively absorbs and corrects small prediction errors, preventing them from cascading into subsequent steps. This greatly enhances the stability and consistency of generation over long sequences. **Nevertheless, traditional discrete visual tokenizers are typically pre-trained and decoupled from downstream task training. While they benefit from the error isolation properties of discrete tokens, their representational capacity is often suboptimal for complex tasks such as multimodal understanding and generation. This creates a visual bottleneck that limits the overall performance of the system.**
>
> To address this, we propose End-to-End Vision Tokenizer Tuning, a method that jointly optimizes the visual tokenizer and the large language model in an end-to-end fashion. **ETT bridges the gap between the tokenizer and downstream training, allowing discrete tokens to retain the stability benefits of the discrete paradigm while gaining richer semantic capacity better suited for multimodal downstream tasks. By resolving the non-differentiability problem of traditional VQ index outputs, we enable effective gradient backpropagation through the tokenizer to substantially enhance the representational power of discrete tokens.** As we emphasize in the main text, our ETT method is specifically designed to address the visual bottleneck issue in MLLMs that use discrete visual tokenizers. In contrast, continuous visual encoders, such as those used in Qwen-VL or LLaVA, have already seen relatively mature development. **Therefore, optimizing continuous visual representations is beyond the scope of our current work.** As shown in Table 2 of the main paper, our method achieves comparable performance to existing MLLMs based on continuous vision encoders, despite using significantly smaller model sizes and less training data. This further demonstrates the effectiveness of ETT in closing the performance gap between discrete and continuous MLLMs, and highlights its potential to unlock further improvements for discrete-token-based models under the NTP paradigm.
>
>
> > [**Q3**]. Questionable Need for Joint Objectives: Table 5 suggests limited synergy between visual reconstruction and multimodal understanding. In fact, optimizing for one often comes at the expense of the other. This raises the question of whether it is necessary, or even desirable, to jointly optimize for both tasks within a single model. The paper does not justify why a unified approach is preferable over specialized models optimized independently for understanding or generation.
>
> [**A3**]. We would first like to clarify that the core motivation behind ETT is to enable the proposed end-to-end tuning to enhance the capability of discrete vision tokenizers in serving downstream multimodal understanding and generation tasks. Therefore, our design intentionally integrates both visual reconstruction and multimodal understanding objectives during training. This is essential: **the multimodal understanding objective allows the discrete VQ tokenizer to acquire high-level vision-language alignment semantics required by understanding tasks, while the visual reconstruction objective ensures the preservation of low-level visual details crucial for visual generation tasks.** We appreciate that Reviewer Snt5 has also recognized this motivation and highlighted it positively in their comment: "ETT maintains the original reconstruction performance of the vision tokenizer, which is essential for tasks that rely on high-quality visual outputs". As both your comment and our analysis in the ablation study on Table 5 (lines 257–269) indicate, there exists an inherent trade-off between multimodal perception and visual reconstruction, we have not yet observed any potential synergy between the two objectives. Nonetheless, in order to support a unified representation that is useful for both multimodal understanding and generation, we deliberately select a joint tuning configuration that balances this trade-off and identifies a practical sweet spot.

---

> > ### Comment · Reviewer_Wh8j · 2025-08-03
> >
> > I thank the authors for their thorough response. While I remain unconvinced by the rationale for merging visual understanding and generation into a single framework, I acknowledge that it is an established problem in the field, and the authors' work represents a meaningful advancement in this area. I strongly recommend that the revised paper include results using a jointly trained tokenizer for both understanding and generation as the discrete visual tokenizer candidate, as this would further strengthen the contribution. I am raising my score to 5 and recommend acceptance.

---

> > > ### Author Response · Authors · 2025-08-03
> > > **Sincere Gratitude for Your Generous Recognition and Substantial Rating Improvement of Our Work**
> > >
> > > **Thank you very much for updating your final score towards 5 and recommending acceptance with the valuable comment "I acknowledge that it is an established problem in the field, and the authors' work represents a meaningful advancement in this area". We sincerely appreciate your great recognition of our work and the efforts we have dedicated during the invaluable rebuttal period.** We are truly gratified that our revised manuscript has addressed most of your concerns and convinced you of the solidity of our paper's content. We will release the experimental results that are being supplemented on the way as soon as possible during the discussion period and add them all to the final revised version. If you have any further questions, please do not hesitate to let us know and we will spare no effort to respond promptly.

---

> ### Author Response · Authors · 2025-08-06
> **Additional Experiments with Different LLM Candidates**
>
> Due to the extremely limited time during the discussion period, to demonstrate that ETT is both scalable and model-agnostic, we have promptly supplemented the ablation studies using Vicuna-7B-v1.5 as the LLM candidate on 12M vision-language understanding data. **It is worth emphasizing that, to obtain results as soon as possible, we follow the same training settings used for Qwen2.5-1.5B when conducting experiments with Qwen2.5-3B and Vicuna-7B-v1.5 as LLM candidates. However, we believe that under the ETT architecture, the optimal training settings should vary across different LLMs. Therefore, the actual upper bound of performance gains that ETT can achieve with different LLMs could be even higher.** **Together with the previously reported ablation results using Qwen2.5-3B on the same data, these findings further confirm the strong potential of our proposed ETT approach.** As promised, the remaining experiment--taking one of the jointly trained tokenizers for both understanding and generation as the discrete visual tokenizer candidate--is still on the way. **In the remaining short time, we will prioritize reporting the corresponding performance gains brought by ETT on multimodal understanding tasks.**
>
> **Table 1: Ablation study on the benefits of ETT for multimodal understanding with different LLMs.**
> | ETT | LLM | SEED$^I$ | GQA | TQA | MME$^P$ |
> |:------|:-------:|:----:|:-----------:|:----:|:-----------:|
> |   | Vicuna-7B-v1.5 |  45.3  |  46.2  |  40.7  |  828.5 |
> | √ | Vicuna-7B-v1.5 |  49.5  |  49.5  |  41.5  |  904.0 |

---

> > ### Author Response · Authors · 2025-08-08
> > **Additional Experiments with Different VQ Candidates**
> >
> > Due to the extremely limited time during the discussion period, to demonstrate that ETT is both scalable and model-agnostic, we have promptly supplemented the ablation studies using UniTok as the VQ candidate and understanding-only end-to-end tuning on 12M multimodal understanding data. **Together with the previously reported ablation results using Qwen2.5-3B and Vicuna-7B-v1.5 as LLM candidates on the same data, these findings further confirm the strong potential of our proposed ETT approach. It is worth emphasizing that, to obtain results as soon as possible, we follow the same training settings used for IBQ when conducting experiments with UniTok as VQ candidate. However, we believe that under the ETT architecture, the optimal training settings should vary across different VQs. Therefore, the actual upper bound of performance gains that ETT can achieve with different VQs could be even higher.** At this point, we have relentlessly completed all the additional experiments we promised to provide during the rebuttal period. It is our sincere hope that our thorough explanations and added experiment results to the best of our abilities will contribute to an improved evaluation score.
> >
> > **Table 1: Ablation study on the benefits of ETT for multimodal understanding with different discrete vision tokenizer candidates.**
> > | ETT | VQ | LLM | SEED$^I$ | GQA | TQA | MME$^P$ |
> > |:------|:-----------:|:-----------:|:----:|:-----------:|:----:|:-----------:|
> > |   | UniTok | Qwen2.5-1.5B |  66.7  |  57.5  |  50.6  |  1274.5  |
> > | √ | UniTok | Qwen2.5-1.5B |  69.8  |  60.5  |  57.9  |  1327.7  |

---

### Author Response · Authors · 2025-08-01
**Overall Response**

We thank reviewers for all the valuable feedback, and the positive comments on **meaningful research perspective** (*Reviewer Wh8j*, *Reviewer Snt5*, *Reviewer xzNc*, *Reviewer t4q5*), **potential contributions to the community** (*Reviewer Wh8j*, *Reviewer Snt5*), **good writing** (*Reviewer Snt5*, *Reviewer xzNc*, *Reviewer t4q5*) and **extensive evaluations and comparisons** (*Reviewer Wh8j*, *Reviewer Snt5*, *Reviewer xzNc*, *Reviewer t4q5*).

We address all the reviewers' comments below and have incorporated all feedback in the revised manuscript. **We sincerely aspire that our detailed rebuttal will dispel any uncertainties or misunderstandings which reviewers may have raised regarding our manuscript, thus contributing positively to the final ratings of this work. If any additional experiments are needed to further demonstrate the potential of our ETT, we will do our utmost to supplement the relevant experiments during the valuable discussion period.**

---

### Note · Authors · 2025-08-11

Dear Reviewer t4q5,

**We would like to touch base with you to see whether you had a chance to look at our wholehearted response. We hope that it has helped address all the concerns you have raised in your reviews.**

Thanks again for your valuable time! We sincerely hope that you will find our work deserving of your esteemed recognition and that it may receive a promising final rating.

Best,

Authors

---

### Decision · Program_Chairs · 2025-09-17

**Decision:**

Accept (poster)

**Comment:**

Reviewers initially raised concerns regarding generalizability (reviewer `Wh8j`), experimental comparisons (reviewer `Wh8j`, `Snt5`, `t4q5`), deeper ablation studies (reviewer `xzNc`), and technical contribution (reviewer `xzNc`, `t4q5`). After the rebuttal, the authors successfully addressed these issues by clarifying their methodology and adding comparisons with relevant baselines. As a result, all reviewers updated their recommendations to clear accept. The authors are suggested to add the essential ablation studies and experiments in the revision.

The paper makes contribution to the visual tokenizer for unified generation and understanding model. The proposed method demonstrates strong empirical performance across diverse benchmarks and highlights a promising direction toward bridging generative and interpretive capabilities.

Given the significance of the contribution, the satisfactory resolution of reviewer concerns, and the consensus among reviewers after the rebuttal, I recommend acceptance of this paper.